# Repressing miR-23a promotes the transdifferentiation of pancreatic α cells to β cells via negatively regulating the expression of SDF-1α

Hongmei Lang[1☯], Ning Lin[2☯], Xiaorong Chen[1,3], Jie Xiang[1,3], Xingping Zhang[1]*, Chao Kang[2]*

1 Department of General Medicine, Chengdu Second People's Hospital, Chengdu, Sichuan Province, China, 2 Department of Clinical Nutrition, the General Hospital of Western Theater Command, Chengdu, Sichuan Province, China, 3 College of Medicine of Southwest Jiaotong University, Chengdu, Sichuan Province, China

☯ These authors contributed equally to this work.
* chao.kang_tmmu@hotmail.com (CK); 194955495@qq.com (XZ)

**Data Availability Statement:** The authors declare that the data supporting the findings of this study and relevant source data are available within the article and its Supplementary Information.

## Abstract

Pancreatic β-cell failure is a pathological feature in type 1 diabetes. One promising approach involves inducing transdifferentiation of related pancreatic cell types, specifically α cells that produce glucagon. The chemokine stromal cell-derived factor-1 alpha (SDF-1α) is implicated in pancreatic α-to-β like cell transition. Here, the serum level of SDF-1α was lower in T1D with C-peptide loss, the miR-23a was negatively correlated with SDF-1α. We discovered that exosomal miR-23a, secreted from β cells, functionally downregulates the expression of SDF-1α, leading to increased *Pax4* expression and decreased *Arx* expression *in vivo*. Adenovirus-vectored miR-23a sponge and mimic were constructed to further explored the miR-23a on pancreatic α-to-β like cell transition *in vitro*, which yielded results consistent with our cell-based assays. Suppression of miR-23a upregulated insulin level and downregulated glucagon level in STZ-induced diabetes mice models, effectively promoting α-to-β like cell transition. Our findings highlight miR-23a as a new therapeutic target for regenerating pancreatic β cells from α cells.

## Introduction

Type 1 diabetes (T1D) exerts a significant impact on morbidity and premature mortality, particularly among young individuals. It is characterized by the destruction of insulin-producing β cells in the pancreatic islets of Langerhans as outlined in the Eisenbarth model [1]. This model depicts the variation in β-cell mass across different age stages, elucidating a chronological sequence that commences with underlying genetic predisposition, followed by an environmental trigger leading to islet-specific autoimmunity, subsequent β-cell loss, dysglycemia, clinical expression of diabetes, and an accelerated progression towards complete β-cell loss [2]. Recent studies have revealed that, following pancreatic β-cell extreme loss, pancreatic α-cells

**Funding:** This research was funded by the Science and Technology Department of Sichuan Province (No. 2021YJ0209), and the Health and Family Planning Commission of Sichuan Province (No. 21PJ070). The funders had no role in study design, data collection and analysis, decision to publish, or preparation of the manuscript.

**Competing interests:** The authors have declared that no competing interests exist.

can transform into β-cells, while the mechanism remains unclear [3]. Aristaless related homeobox (Arx), and Paired box gene (Pax) 4 play pivotal roles in directing the differentiation of pancreatic precursor cells into β cells and α cells, respectively [4]. Suppression of Arx expression induces differentiation of precursor cells into β cells [5]. Genetic studies showed that ectopic expression of Pax4 or inactivation of Arx in cells resulted in the neogenesis of functional β cells from α cells [6, 7]. These findings provide evidence that the fate of islet sub-types primarily relies on the reciprocal repression between Pax4 and Arx factors.

The chemokine, stromal cell-derived factor-1 (SDF-1α), known as CXCL12, represents an attractive therapeutic target molecule in T1D. Among African Americans with type 1 diabetes, those in the lowest quartile of plasma SDF-1 (under 985.0 pg/mL) exhibited the highest risk of mortality compared to others [8]. SDF-1α signaling plays a crucial role in β-cell development, survival and regeneration [9]. Transgenic mice overexpressing SDF-1α within their β-cells are resistant to streptozotocin (STZ)-induced β cell apoptosis and diabetes [10]. Following an injury to β cells, SDF-1α signaling is postulated to facilitate α-cell hyperplasia and subsequent trans-differentiation into β cells, shedding light on the involvement of SDF-1α in the transition from α to β cells [11].

MicroRNAs (miRNAs) are a type of non-coding RNA with a length of about 22 nucleotides and a relatively conservative sequence. Recent studies have demonstrated their significance in pancreatic development, islet regeneration, the regulation of islet function, and their association with of type 1 diabetes [12]. It is not known whether miRNA could manipulate SDF-1α. In this study, we explored four databases to pick out five miRNAs that predicted to interact with SDF-1α. miR-23a was chosen through validation in T1D patients with C-peptide under 200 pmol/L. To explore the molecular mechanism of miRNA on the regulation of pancreatic islet function, primary islet cells were utilized to confirm the inhibitory effect of miR-23a on SDF-1α. Exosomes derived from MIN6 cells were co-cultured with pancreatic α cell lines to validate the role of miR-23a in α-to-β like cell transition. Additionally, we intraperitoneally injected adenovirus-vectored miR-23a mimic or sponge into STZ-induced diabetes mice to investigate the impact of miR-23a on α-to-β like cell transition. To our knowledge, our results reveal a new mechanism of SDF-1α in pancreatic α-to-β cell transdifferentiation and suggest that miR-23a is a potential biomarker of severe T1D.

## Materials and methods

### Human study

Samples of fasting venous blood were obtained from 18 individuals in the General Hospital of Western Theater Command from May to December 2022. The entire cohort was equally sub-divided into subgroups of healthy and T1D individuals with or without C-peptide under 200 pmol/L. The T1D patients who used dipeptidyl peptidase 4 (DPP-4) inhibitor were excluded, because SDF-1α was easily affected by DPP-4 inhibitor [13]. Blood was centrifuged at 3,000 rpm for 15 minutes at 4°C to remove whole cells, cell debris, and aggregates. The metabolic parameters of blood, including fasting blood glucose (FBG), HbA1c, C-peptide and fasting insulin were tested by the Department of Laboratory Medicine of the General Hospital of Western Theater Command. SDF-1α was measured by using an enzyme-linked immunosorbent assay kit (Human CXCL12/SDF-1α immunoassay, R&D Systems, Minneapolis, USA). Here, 200 pmol/L of plasma C-peptide was chosen as the cut-off for severe insulin deficiency or β cell impairment [14]. All subjects gave written informed consent before blood collection. The study was conducted according to the principles of the Declaration of Helsinki and approved by the Ethics Committee of the General Hospital of Western Theater Command.

The clinical trial was registered at http://www.chictr.org/cn/ under study number ChiCTR2200059448.

## RNA isolation and quantitative RT-PCR

RNA isolation (RNAeasy, QIAGEN) and cDNA synthesis (Superscript choice system, Invitrogen) was performed according to the manufacturer's instructions. Quantitative RT–PCR was carried out using the QuantiTect SYBR Green RT-PCR Kit (QIAGEN) and validated primers (QIAGEN) according to the manufacturer's instructions. The PCR reactions and detection were performed on a light cycler (Roche) using GAPDH as internal controls for normalization purposes. The corresponding PCR primers were provided in S1 Table.

## The miRNA target prediction

The miRNA target prediction and analysis were performed with the algorithms from TargetScan (http://www.targetscan.org/), PicTar (http://pictar.mdc-berlin.de/), miRanda (http://www.microrna.org/) and microT (https://micro-t-software.software.informer.com/).

## Luciferase assay

The reporter plasmid p-MIR-SDF-1α containing the predicted miR-23a targeting regions was designed by Genescript (Nanjing, China). Part of the wild-type and mutated 3'-UTR of SDF-1α was cloned immediately downstream of the firefly luciferase reporter. The 2 mg of β-galactosidase expression vector (Ambion) was used as a transfection control. For the subsequent luciferase reporter assays, 2 mg of firefly luciferase reporter plasmid, 2 mg of β-galactosidase vector, and equal doses (200 pmol) of mimics, inhibitors or scrambled negative control RNA were transfected into the prepared cells. At 24 h after transfection, cells were analyzed using the Dual Luciferase Assay Kit (Promega) according to the manufacturer's instructions. Each sample was prepared in triplicate and the entire experiment was repeated three times.

## Isolation of exosomes from medium

Exosomes in the medium were isolated from the cell by differential centrifugation, according to previous research [15]. After removing cells and other debris by centrifugation at 300 g and 3,000 g respectively, the supernatant was then centrifuged at 10,000 g for 30 min to remove shedding vesicles and the other vesicles with larger sizes. Finally, the supernatant was centrifuged at 110,000 g for 70 min, and exosomes were collected from the pellet and re-suspended in PBS (all steps were performed at 4°C).

## Islet isolation, dispersion and culture

Animals were sacrificed by cervical dislocation and the pancreas was perfused with 2 ml of 900 U/ml collagenase (Sigma Aldrich, USA) in HBSS (Life Technologies, USA). After surgical removal of the pancreas, it was digested in 2 ml of collagenase solution at 37°C for 13 min, followed by manual shaking for 60–90s, two rounds of washing, and passing through a 70 μm filter. Islets were hand-picked and cultured in RPMI 1640 (PAN-Biotech, Germany) supplemented with 2 mM L-Glutamine, 100 U/ml penicillin, 100 mg/ml streptomycin, and 10% FBS (Gibco 10270–106). Islets were recovered overnight before secretion assays or dispersion. For subcellular localization studies, islets were dispersed into single cells by pipetting in 0.05% trypsin–EDTA (Gibco, USA) solution for 1 min, seeded onto uncoated glass coverslips, and cultured for six days prior to fixation.

## Cell culture

αTCl-6 cells were purchased from ATCC (CRL-2934) and grown in low-glucose DMEM supplemented with 10% FBS, 50 U/mL penicillin and 50 mg/mL streptomycin. Min6 cells with doxycycline-inducible constructs were grown in high-glucose DMEM supplemented with15% Tet System Approved FBS (Clonetech 631106), 71 mM 2-mercaptoethanol, 50 U/ml penicillin and 50 mg/ml streptomycin. The mouse islets from the cell lineage-tracing animals were kept in RPMI medium supplemented with 10% FBS, 50 U/ml penicillin and 50 mg/ml streptomycin. The cell culture for human islets followed established protocols.

## Electron microscopy

For immunogold staining, 200 islets, isolated by collagenase (1mg/ml) digestion, were fixed with 4% paraformaldehyde, and 0.2% glutaraldehyde in 0.1M phosphate buffer (PB) (pH 7.4) overnight at 4˚C and were processed for ultracryomicrotomy according to a slightly modified Tokuyasu method (Tokuyasu, 1973). In brief, islets were spun down in 10% gelatin. After immersion in 2.3 M sucrose (in [pH 7.4], 0.1M PB) overnight at 4˚C, the samples were rapidly frozen in liquid nitrogen. Ultrathin (70 nm thick) cryosections were prepared with an ultra-cryomicrotome (Leica EMFCS, Austria) and mounted on formvar-coated nickel grids (Electron Microscopy Sciences, Fort Washington, PA, USA). Immunostainings were processed with an automated immunogold labeling system Leica EM IGL as follows: the grids were incubated successively in PBS containing 50mM $NH_4Cl$, PBS containing 1% BSA, PBS containing both anti-insulin and anti-glucagon primary antibodies diluted 1/1000 in 1% BSA for 1h, PBS containing 0.1% BSA, PBS containing 1% BSA and both 10 nm and 15 nm colloidal gold conjugated anti-guinea pig IgG and anti-mouse IgG, respectively, (BBInternational, Cardiff, UK), PBS containing 0.1% BSA for 5min, PBS for 5min twice. Lastly, the samples were fixed for 10 min with 1% glutaraldehyde, rinsed in distilled water and contrasted with a mixture of 1.8% methylcellulose and 0.3% uranyl acetate on ice. After air-drying, sections were examined under a JEOL 1400 transmission electron microscope.

## Transmission electron microscopy assay (TEM)

The exosome pellet was placed in a droplet of 2.5% glutaraldehyde in PBS buffer at pH 7.2 and fixed overnight at 4˚C. Samples were rinsed in PBS buffer and post-fixed in 1% osmium tetroxide for 60 min at room temperature. The samples were then embedded in 10% gelatin and fixed in glutaraldehyde at 4˚C and cut into several blocks (less than 1 $mm^3$). The samples were dehydrated for 10 min each step in increasing concentrations of alcohol (30, 50, 70, 90, 95, and 100% × 3). Pure alcohol was then exchanged by propylene oxide, and specimens were infiltrated with increasing concentrations (25, 50, 75, and 100%) of Quetol-812 epoxy resin mixed with propylene oxide for a minimum of 3 h per step. Samples were embedded in pure, fresh Quetol-812 epoxy resin and polymerized at 35˚C for 12 h, 45˚C for 12 h, and 60˚C for 24 h. Ultrathin sections (100 nm) were cut using a Leica UC6 ultra-microtome and post-stained with uranyl acetate for 10 min and with lead citrate for 5 min at room temperature before observation in an FEI Tecnai T20 transmission electron microscope, operated at 120 kV.

## Nanoparticle tracking analysis

The size and density of exosomes were directly tracked using the Nanosight NS 300 system (NanoSight technology, Malvern, UK) [16]. Exosomes were re-suspended in PBS at a concentration of 5 μg/mL, and were further diluted 100- to 500-fold to achieve between 20 and 100 objects per frame. Samples were manually injected into the sample chamber at ambient

temperature. Each sample was configured with a 488 nm laser and a high-sensitivity sCMOS camera and was measured in triplicate at camera setting 13 with an acquisition time of 30s and a detection threshold setting of 7. At least 200 completed tracks were analyzed per video. Finally, data were analyzed using the NTA analytical software (version 2.3).

## Western blotting

The SDF-1α, Arx, Pax4, Insulin and Glucagon expression were assessed by western blotting analysis and samples were normalized to GAPDH. Protein extraction was blocked with PBS-5% fat-free dried milk at room temperature for 1 h and incubated at 4˚C overnight with anti-SDF-1α (1:1000, Santa cruz), anti-Arx (1:1000, Santa cruz), anti-Pax4 (1:1000, Santa Cruz), anti-Insulin (1:1000, Santa cruz), anti-Glucagon (1:1000, Santa cruz), anti-Aldh1a3 (1:1000, Novus), anti-Neurog3 (1:1000, Beta Cell Biology Consortium), anti-MafA (1:1000, Cell Signaling Technology), anti-Pdx1 (1:1000, Cell Signaling Technology), anti-NeuroD1 (1:1000, Cell Signaling Technology) and anti-XBP1 (1:1000, Santa cruz), anti-CD63 (1:2000, Abcam), anti-TSG101 (1:1000, Santa Cruz), anti-Ago2 (1: 1000, Santa Cruz) and anti-GAPDH (1:3000, Santa Cruz) antibodies respectively.

## Adenovirus constructs

The miR-23a mimic adenovirus and miR-23a sponge adenovirus used in animal experiments were purchased from Jima Pharmaceutical Technology Co., Ltd, Shanghai, China. The adenovirus was amplified from 293A cells. When the cells grew to approximately 50% confluence, the 293A cells were added to the adenovirus seed. After culturing for 2–3 days, a cytopathic effect (CPE) was observed. Following this, adenovirus stock was harvested through repeated freezing and thawing, and it was purified using the ViraTrapTM Adenovirus Purification Miniprep Kit (Biomiga, San Diego, USA).

## Animal maintenance and manipulations

Male C57BL6J mice were housed and used according to the guidelines of the Belgian Regulations for Animal Care, with the approval of the Ethical Committee at the general hospital of Western Threaten Command. To develop a T1D model, mice were fasted overnight, but allowed free access to water, prior to injection of streptozocin (STZ, Sigma-Aldrich). STZ solution was prepared in citrate buffer (0.5%, pH = 4.3). Mice were given by intraperitoneal injection at a dosage 40 mg/kg of body weight for five consecutive days. Meanwhile, healthy control mice were injected with the same volume of sodium citrate. After 3 days of STZ injection, blood glucose level was measured from a tail nick using a handheld glucometer (ACCU-CHEK Active, Mannheim, Germany). The mice in the sham group were injected with the same volume of sodium citrate solution. After two consecutive measurements, mice were considered as T1D when random blood glucose exceeded 200 mg/dL. The mice were then fed with a regular diet. STZ-mice were intraperitoneally injected with Adeno Associated Virus (AAV) serotype 9 expressing miR-23a sponge (AAV9-miR-23a sponge, n = 10, $1\times10^{11}$VP/mouse, once a week) or miR-23a mimic (AAV9-miR-23a mimics) construct for 6 weeks. The miR-23a mimic adenovirus and inhibitor vector adenovirus used in the animal experiments were purchased from Shanghai Jima Pharmaceutical Technology Co., Ltd. The pancreatic tissues were immediately stored at -80˚C for further analysis. All experiments were performed in accordance with relevant regulations and the ARRIVE (Animal Research: Reporting on In Vivo Experiments) guideline.

## Oral glucose tolerance test (OGTT)

Mice were fasted overnight (14 h) before glucose tolerance tests and glucose (2.0 g/kg body weight) was administered by oral gavage. Blood glucose levels were measured from tail bleeds with a glucometer (Roche, Accu-Chek Performa) at 0, 15, 30, 60, 90 and 120 min. Glucose tolerance was assessed by calculating the incremental area under the curve (AUC) of each GTT.

## HE and IHC

The tissues were fixed in 4% paraformaldehyde, embedded in paraffin, sectioned and then stained with corresponding antibodies (Abcam). Quantitative analysis was conducted by quantifying the fluorescence intensity from at least five sections. β-cell mass analysis was calculated using the Aperio Spectrum™ software. β-cell area was calculated as the percentage of the entire pancreatic section staining positive for insulin. β-cell mass was calculated as the product of the β-cell area and pancreas weight.

## Statistical analyses

Data were analyzed using Prism software (GraphPad) by first determining whether they followed a normal distribution. The variables were compared using Fisher's exact test and Chi-square test for categorical outcomes. An unpaired t-test (2 groups being compared) or an unpaired ANOVA test (several groups compared simultaneously) was used assuming Gaussian distribution. All statistical tests were 2-tailed and, unless otherwise noted, $P<0.05$ was considered the level of statistical significance. Variables are expressed as means (SD).

# Results

## Serum miR-23a was negatively correlated with SDF-1α in T1D patients

The serum level of SDF-1α in individuals with T1D was initially measured, and it exhibited a decline upon C-peptide loss (Table 1). To identify the microRNAs which predicted to combine with the target gene of SDF-1α, we found common five microRNAs (hsa-miR-23a-3p, hsa-miR-23b-3p, hsa-miR-141-3p, hsa-miR-144-3p and hsa-miR-200a-3p) in four databases (PicTar, TargetScan, microT and miRanda) (Fig 1A). Notably, miR-23a exhibited a significant increase in individuals with T1D, particularly those with C-peptide loss (Fig 1B). Furthermore, a negative correlation between miR-23a and serum SDF-1α was observed exclusively in T1D with C-peptide under 200 pmol/L (Fig 1C–1E).

**Table 1. Clinical parameters of T1D.**

| Variable | Healthy control (n = 6) | T1D (n = 6) | T1D with C-peptide under 200 pmol/L (n = 6) | P Value |
|---|---|---|---|---|
| Age (years) | 28.00±5.29 | 22.83±5.56 | 26.83±6.49 | 0.300 |
| Gender: men, No. (%) | 3 (50.0) | 4(66.6) | 3 (50.0) | |
| FBG (mmol/L) | 5.20±0.59 | 10.08±0.82 | 11.88±1.79 | <0.001 |
| HBA1C (%) | 5.73±0.31 | 7.05±1.33 | 8.72±0.94 | <0.001 |
| C-peptide (pmol/L) | 1728.33±550.94 | 1343.33±307.09 | 162.50±52.52 | <0.001 |
| SDF-1α (pg/mL) | 3048.64±525.09 | 2538.99±427.64 | 1877.19±336.59 | 0.001 |

Data are presented as mean±SD. Significance was analyzed by One-way ANOVA. Abbreviations: FBG, Fasting blood glucose; HbA1C, glycosylated hemoglobin.

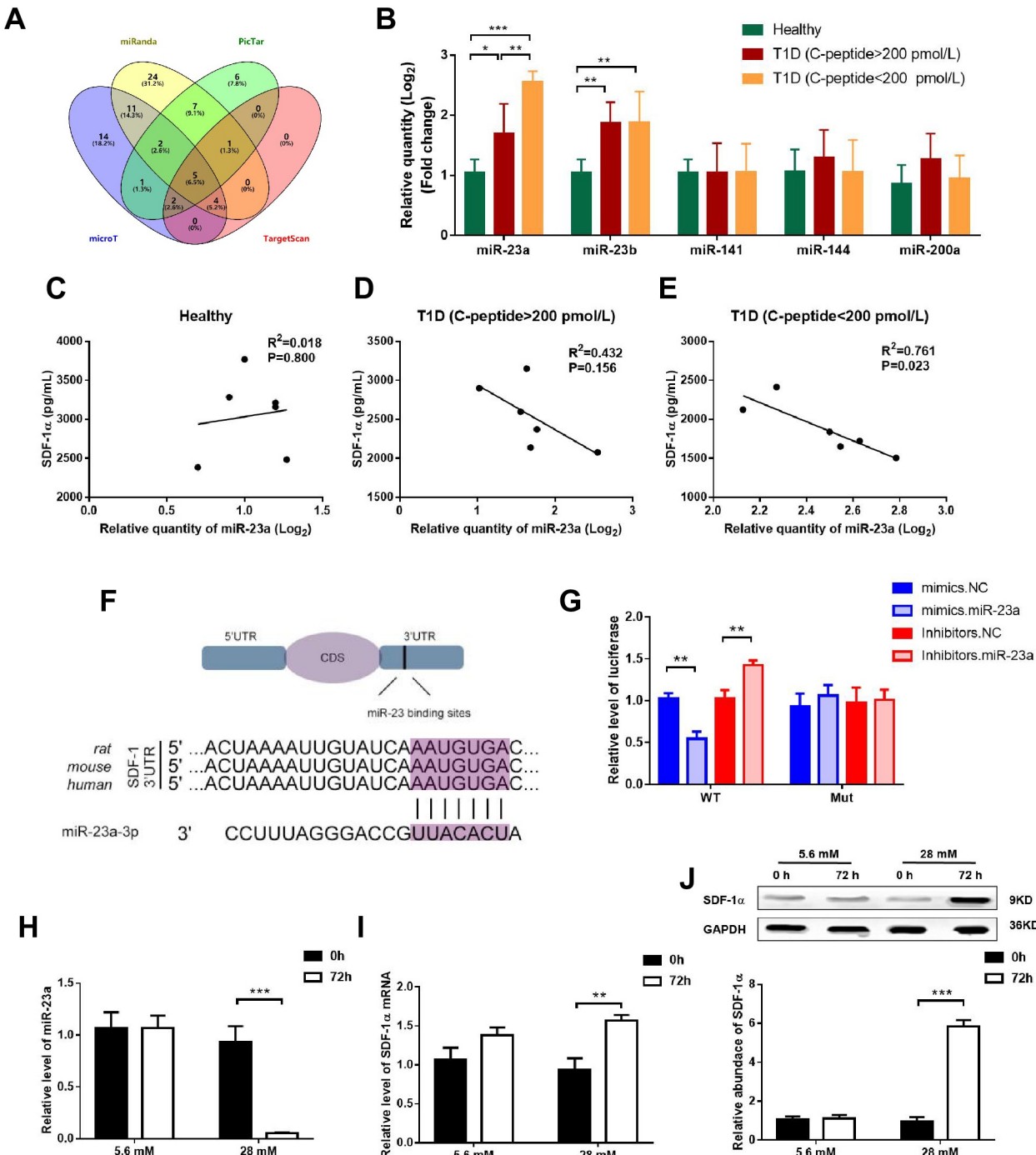

**Fig 1. miR-23a negative regulates the expression of SDF-1α.** (A) Venn diagram of the predicted microRNA of targeted SDF-1α in PicTar, TargetScan, microT and miRanda database. (B) Serum circulating predicted microRNA of targeted SDF-1α. miRNAs are ordered based on log2 fold change. (C-E) Correlation between miR-23a and SDF-1α in healthy subjects and T1D patients with or without C-peptide under 200 pmol/L. (F) The binding sites of miR-23a in the 3'-UTR of SDF-1α mRNA. Predicted binding sites of miR-23a within the 3'-UTR of SDF-1α mRNA. (G) Direct recognition of SDF-1α 3'-UTR by miR-23a. Primary islet cells were co-transfected with firefly luciferase reporters containing either wild-type or mutant (mut) SDF-1α 3'-UTR with miR-23a mimics, inhibitors and the corresponding normal control. The relative luciferase levels were detected using a luciferase kit at 24–36 h after transfection (n = 3). (H) High glucose inhibited the expression of miR-23a in primary islet cells. (I-J) High glucose promoted the mRNA and Protein expression of SDF-1α in primary islet cells. The data represent the mean±SD. *P < 0.05, **P < 0.01, ***P < 0.001.

## miR-23a directly targeted SDF-1α and negatively regulated α-to-β like cell transition *in vitro*

To give direct evidence of the interaction between miR-23a and SDF-1α, we employed luciferase reporter plasmid containing either wild-type or mutant 3'-UTR of SDF-1α mRNA; the binding sites of miR-23a were shown in Fig 1F. Our findings demonstrated a significant reduction in luciferase activity in cells overexpressing miR-23a, whereas inhibition miR-23a resulted in a relative enhancement of luciferase activity (Fig 1G). Importantly, the inhibitory effect of miR-23a on luciferase activity was abolished upon the loss of the binding sites (Fig 1G). These results identified SDF-1α as a direct target of miR-23a.

To examine the essential role of miR-23a in α-to-β like cell transition *in vitro*, primary islet cells were exposed to glucose concentrations of 5.6 mmol/l and 28 mmol/l for 72 hours, mimicking the hyperglycemic conditions characteristic of diabetes and effectively replicating the state of chronic hyperglycemia, as previously described [17]. As expected, miR-23a was repressed in 28 mmol/l glucose for 72h incubation (Fig 1H). Additionally, both the mRNA and protein levels of SDF-1α exhibited upregulation under high glucose conditions (Fig 1I and 1J).

## Exosomes miR-23a secreted from MIN6 cells inhibited α-to-β like cell transition *in vitro*

Circulating microRNAs were mainly encapsulated in exosomes [18]. We next isolated the exosomes secreted from both MIN6 cells and αTCl-6 cells (Fig 2A and 2B). The level of miR-23a in both MIN6 cells and exosomes were strongly higher than in αTCl-6 cells and exosomes (Fig 2C), indicating that the miR-23a was primarily secreted by pancreatic β cells. The level of insulin was increased, whereas glucagon was decreased in αTCl-6 cells co-cultured with MIN6 exosomes miR-23a deleted (Fig 2D). Repression of exosomal miR-23a upregulated the protein level of SDF-1α, as well as changed in Arx and Pax4 (Fig 2E). To further confirm the β-cell identity during α-to-β cell transdifferentiation of αTCl-6 cells, we examined some key β-cell enriched transcription factors, such as Insulin, MafA, Pdx1, NeuroD1 and XBP1 at protein levels (Fig 2F). Immunoblotting analysis displayed that these factors were increased after treatment with MIN6 exosomes miR-23a deleted. Accordingly, exosomal miR-23a derepressed α cell markers (glucagon, Arx).

To conclude, exosomal miR-23a secreted from MIN6 cells suppressed the SDF-1α expression, as well as the expression of related transcription factors such as Arx and Pax4 in αTCl-6 cells, thus inhibiting the transdifferentiation of pancreatic α cells to β cells *in vitro*.

## Repression of miR-23a promoted α-to-β like cell transition *in vivo*

To further explore the role of miR-23a in α-to-β like cell transition, an adenovirus-vectored miR-23a sponge was constructed and intraperitoneally injected into STZ-induced diabetes mice for six weeks. STZ-induced mice intraperitoneally injected with AAV9-miR-23a mimics, exhibited the smallest islet area and reduction of β-cell mass as compared to other groups because of β cells loss (Fig 3A and 3B). The body weight showed no significant different among groups (S1 Fig). Inhibition of miR-23a remission alleviated glucose tolerance impairments (Fig 3C and S2 Fig), upregulated the secretion of insulin and decreased glucagon both in the serum and at protein levels (Fig 3D and 3E). As expected, the expression of miR-23a showed a lower level in STZ-induced mice intraperitoneally injected with AAV9-miR-23a sponge than AAV9-miR-23a mimics (Fig 3F). The mRNA and protein levels of SDF-1α were increased in the AAV9-miR-23a sponge group, resulting in corresponding changes in Arx and Pax4 (Fig 3G). Further more, miR-23a deactivated beta cell identity genes (insulin, MafA,

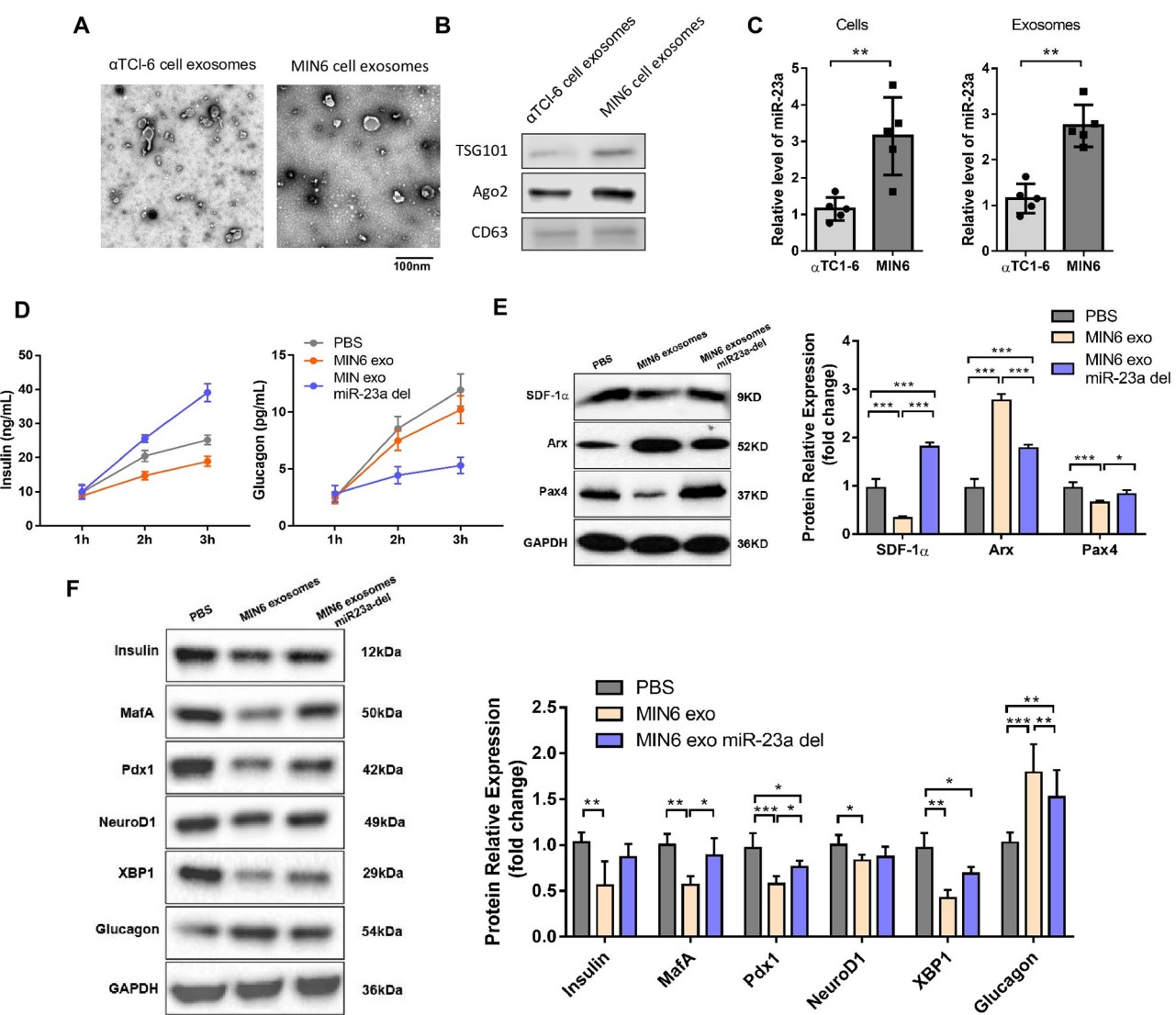

**Fig 2. MIN6 secrete exo-miR-23a to suppress α-to-β transition of αTCl-6 cells.** (A) TEM images of exosomes isolated from MIN6 and αTCl-6 cells (scale bar, 100 nm). (B) WB analysis of exosomal markers in the exosomes isolated from MIN6 and αTCl-6 cells. (C) Quantification of miR-23a in both cells and exosomes. (D) Effects of MIN6 exosomes and control exosomes on the insulin and glucagon secretion from αTCl-6 cells. (E) Effects of MIN6 exosomes and control exosomes on the protein expression of SDF-1, Arx and Pax4 from αTCl-6 cells. (F) α-cell identity markers (glucagon) and β-cell identity markers (Insulin, MafA, Pdx1, NeuroD1 and XBP1). *P < 0.05, **P < 0.01, ***P < 0.001.

Pdx1, NeuroD1, Urocortin, Ero1lβ, XBP1, Fig 3H), while derepressing β-cell dedifferentiation genes (Aldh1a3 and Neurog3, Fig 3I) and α cell genes (glucagon, Fig 3J). In conclusion, repression of exosomal miR-23a in pancreatic β cells increased the expression of SDF-1α in pancreatic α cells, subsequently leading to the upregulation of Pax4 and the downregulation of Arx, ultimately promoting the transdifferentiation of pancreatic α cells into β cells (Fig 4).

## Discussion

Eisenbarth model postulated the change of pancreatic β cell mass alongside with T1D, and restoration of β-cell mass is the leading strategy for T1D. Regeneration of β cells from other cells, especially α cells shows a promising approach [19]. Recent studies have shown that SDF-1α

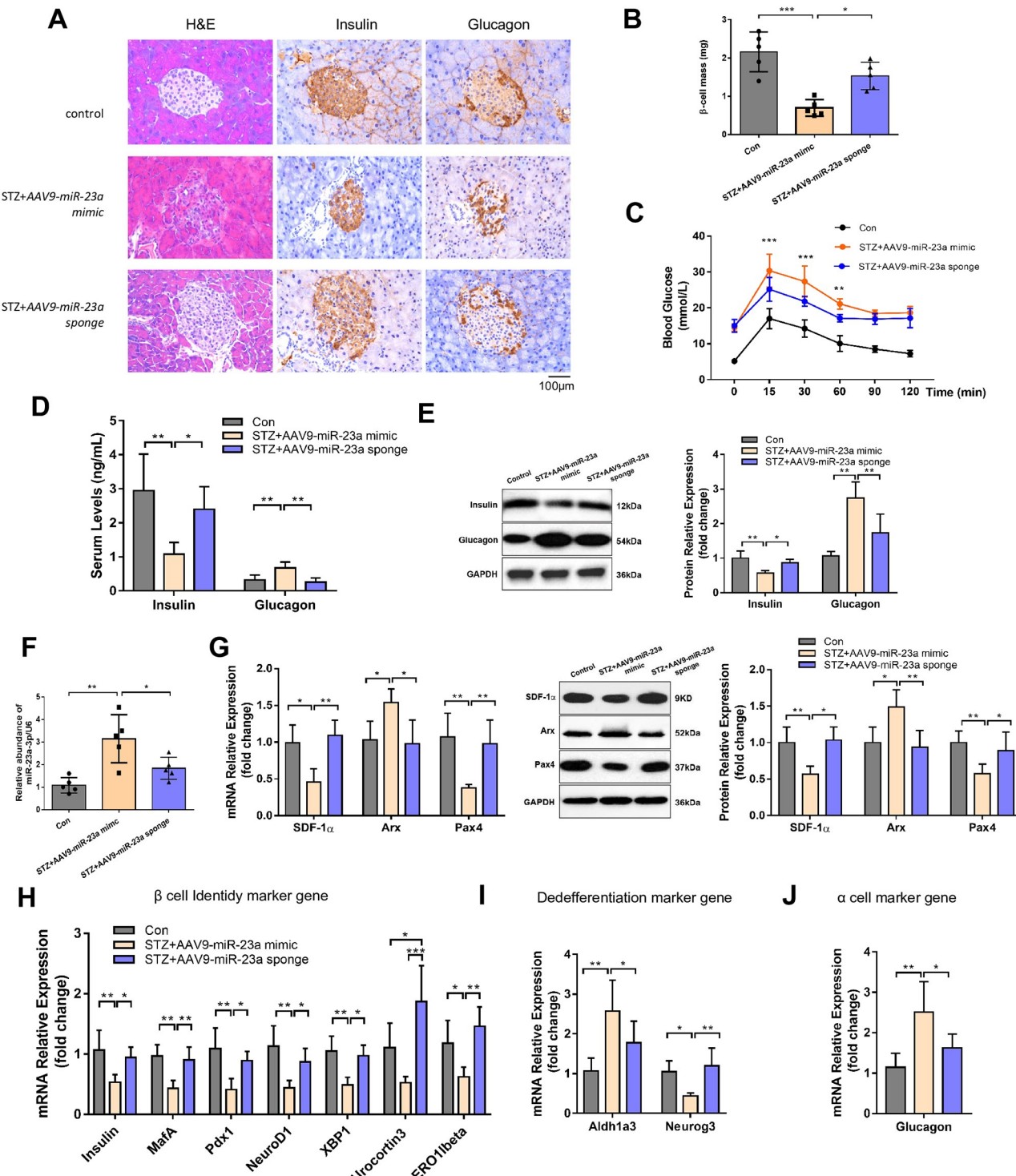

**Fig 3. Repressing miR-23a alleviates diabetes.** (A) H&E staining in the pancreatic (O) of mice was determined. Scale bar: 100μm. (B) β-cell mass. (C) Glucose tolerance. (D-E) Serum levels and protein levels of insulin and glucagon. (F) Gene expression of miR-23a. (G) Gene expression and protein levels of SDF-1α, Arx and Pax4 among groups. (H) Gene expression of β-cell identity markers (insulin, MafA, Pdx1, NeuroD1, Urocortin, Ero1lβ, XBP1). (I) Gene expression of β-cell dedifferentiation markers (Aldh1a3 and Neurog3). (J) Gene expression of α-cell identity marker (glucagon). *P < 0.05, **P < 0.01, ***P < 0.001.

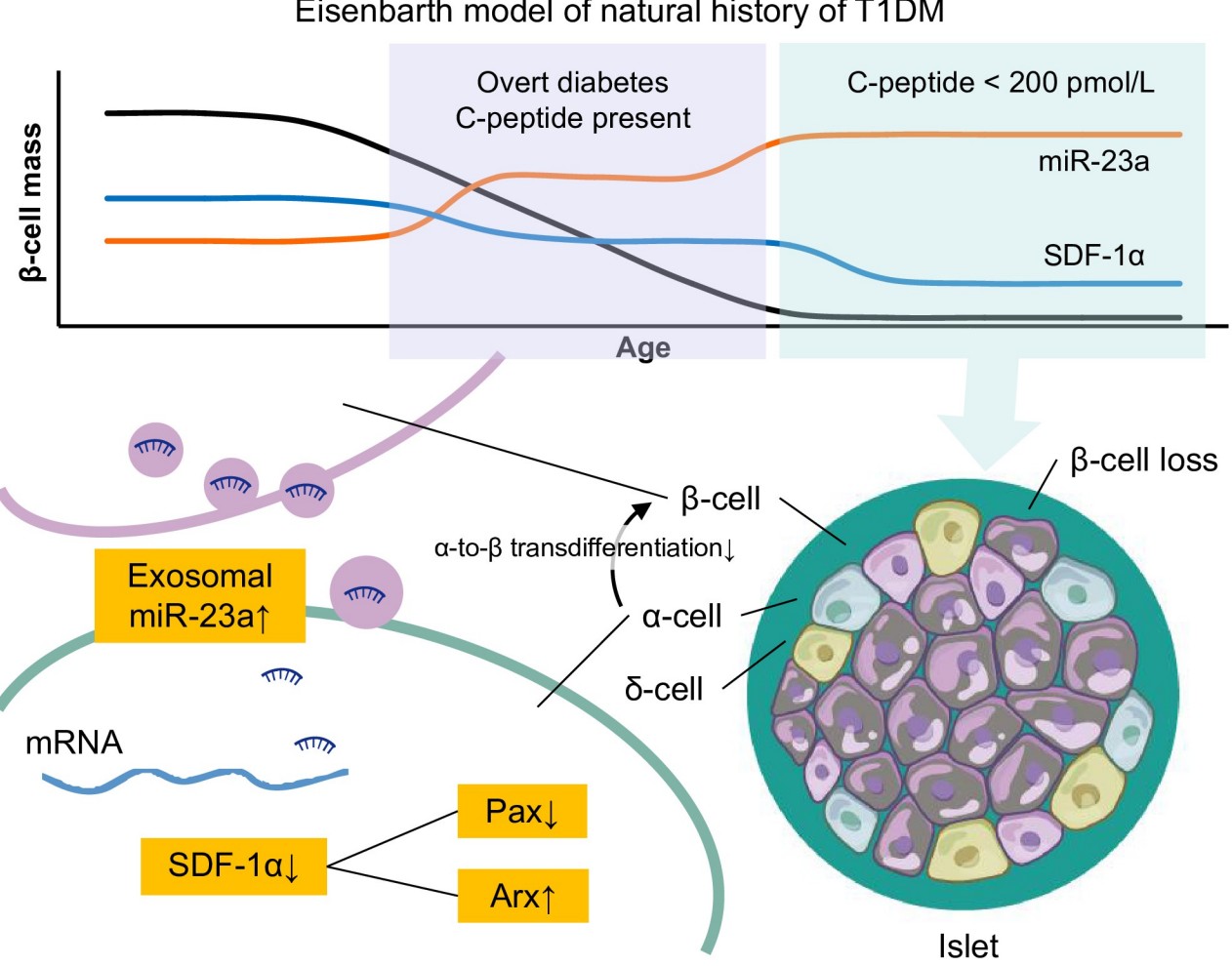

**Fig 4. Possible mechanism of miR-23a on SDF-1 inhibiting the α-to-β like cell transition.**

mediates the transdifferentiation of α cells to β cells, our study showed that miR-23a was directly targeted with SDF-1α. Repression of exosomal miR-23a secreted from β cells upregulated the expression of SDF-1α and promoted the pancreatic α-to-β like cell transition both *in vivo* and *in vitro* (Fig 4). Therefore, inhibition of miR-23a may represent a viable target for the treatment of severe type 1 diabetes.

SDF-1α is expressed at a very low level, which is considered so-called β-cell or islet core disallowed or forbidden genes [20]. SDF-1α (also known as CXCL12-α) signaling is demonstrated to promote β-cell development, survival and regeneration in the pancreatic islets, which is an promising therapeutic target molecule in T1D [21]. Our results showed that the serum levels of SDF-1α were lower in T1D, especially in those with C-peptide loss. Previous research showed that the impaired retention of the Treg cells in non-obese diabetic (NOD) mice correlated with the most prominent decrease observed in SDF-1, whereas induction of allogeneic chimerism recovered the Treg cell population with improved function of the CXCR4/SDF-1 axis in the pancreatic lymph nodes of the NOD [22]. SDF-1α transgenic mice regenerate 50% of pancreatic β cells after 2 weeks, while wild-type mice treated with the same treatment are almost all α cells. Regeneration of pancreatic β cells after injury may be related

to the upregulation of SDF-1α in pancreas, which indicates the transdifferentiation of α cells to β cells [10]. Liu et al. showed that the expression of SDF-1α was increased after pancreatic β-cell injury, and SDF-1α induced the secretion of glucagon-like peptide-1 (Glucagon-like peptide-1, GLP-1) by binding to CXCR4 in pancreatic α-cells [23]. Therefore, SDF-1α represents a promising therapeutic target for T1D.

Our study compared five predicted serum miRNAs of SDF-1α in four miRNA databases, miR-23a was the most abundant in T1D with C-peptide loss and negatively correlated with serum SDF-1α. By using bioinformatics analysis, miR-23a was a directly target of SDF-1α, indicating repression miR-23a may play an important role in T1D. As expected, miR-23a was elevated in T1D with C-peptide over 200 pmol/L. Previous clinical trials showed that plasma circulating miR-23~27~24 clusters correlate with the immune metabolic derangement and predict C-peptide loss in children with type 1 diabetes [24]. miR-23a maybe a biomarker of severe T1D. In recent years, exosomes have been a provocative topic in both the biomarkers of diabetic complications and the therapeutic target of diabetic [25]. These cell-derived small particles are also used as a safe vehicle for the delivery of targeted drugs, as well as miRNAs [26]. Recent studies have found that vesicle miRNA secreted by pancreatic β-cells after apoptosis can be taken up by neighboring β-cells as exocrine bodies, leading to apoptosis of neighboring β-cells, indicating that vesicle miRNA secreted by β-cells can affect the function and activity of recipient cells [27]. Here, we extracted exosomes secreted by both MIN6 cells and αTCl-6 cells to detect the activity of miR-23a, and co-culture exosomes secreted from MIN6 cells with αTCl-6 cells to further explore the mechanism of miR-23a on the transdifferentiation of pancreatic α cells to β cells. Our results showed that miR-23a negatively regulates the expression of SDF-1α, followed by downregulation of Arx and upregulation of Pax4, the marker of α-cell and β-cell identity gene were also changed accordingly.

One inevitable question arises concerning why α-to-β like cell transition does not appear in T1D patients with C-peptide loss, whereas high-glucose induced the transdifferentiation of pancreatic α cells to β cells *in vitro*, which may seem to be contradicted. One possible mechanism was that β cells were not completely lost in humans with C-peptide under 200 pmol/L. Replication of pre-existing β cells, not transdifferentiation from other cells, is the principal mechanism for replenishing or maintaining β-cell mass in adulthood by genetic lineage tracing experiments during both homeostasis and during injury [28]. The role of miR-23/SDF-1α signaling in the progression of diabetes remains unclear. Further studies should investigate the functional status of human β cells at different stages of T1D. Additionally, the mechanism underlying the miR-23/SDF-1α signaling in cross-repression of PAX4 and Arx needs to elucidated. Previous study showed that DNA methylation was essential for the repression of Arx to maintain β cell identity, and simultaneous inactivation of DNA methyltransferase 1 and Arx lead to the pancreatic α-to-β cell transdifferentiation [29]. The relationship between miR-23/SDF-1α signaling and epigenetic foctors involved in the conversion of α to β-like cells should be investigated in the future.

Alpha cells are normal in patients with type 1 diabetes, and there are more α cells in patients with type 2 diabetes. Therefore, transdifferentiation of α cells to β cells provides a promising therapeutic way for T1D. Our study found that repression of miR-23a could promote pancreatic α-to-β like cell transition via up-regulating SDF-1α. miR-23a is expected to be a new target to improve insulin insufficiency in T1D patients.

## Supporting information

**S1 Fig. Body weight.**
(PDF)

**S2 Fig. Area under curve.**
(PDF)

**S3 Fig. Original image of western blot analysis.**
(PDF)

**S1 Table. Sequences of primers.**
(DOCX)

## Author Contributions

**Conceptualization:** Xingping Zhang.

**Data curation:** Xiaorong Chen.

**Formal analysis:** Hongmei Lang, Xiaorong Chen, Jie Xiang.

**Methodology:** Xiaorong Chen, Jie Xiang.

**Supervision:** Ning Lin, Xingping Zhang, Chao Kang.

**Visualization:** Jie Xiang.

**Writing – original draft:** Hongmei Lang, Chao Kang.

**Writing – review & editing:** Ning Lin, Xingping Zhang.

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
