## [Decision Letter · Decision Letter 0]

22 Oct 2023

PONE-D-23-30492Repressing miR-23a promotes the transdifferentiation of pancreatic α cells to β cells via negatively regulating the expression of SDF-1α.PLOS ONE

Dear Dr. kang,

Thank you for submitting your manuscript to PLOS ONE. After careful consideration, we feel that it has merit but does not fully meet PLOS ONE’s publication criteria as it currently stands. Therefore, we invite you to submit a revised version of the manuscript that addresses the points raised during the review process.

Your research offers intriguing insights into the potential therapeutic role of miR-23a for Type 1 Diabetes. However, upon thorough review, several concerns have been raised that necessitate further attention and clarification. Key issues include the clarity of blood glucose measurements, justification for specific experimental conditions, the quality and representation of TEM and IHC images, and the comprehensive examination of transdifferentiation markers. Addressing these concerns is imperative to enhance the robustness and credibility of your study. Detailed comments are attached for your perusal.

We look forward to receiving your revised manuscript.

Kind regards,

Syed M. Faisal, Ph.D.

Academic Editor

PLOS ONE

[This research was supported by the Science & Technology Department of Sichuan Province (No. 2021YJ0209), and the Science And Technology Project of the Health Planning Committee of Sichuan (No. 21PJ070).]

 [The funders had no role in study design, data collection and analysis, decision to publish, or preparation of the manuscript.]

7. Please amend either the title on the online submission form (via Edit Submission) or the title in the manuscript so that they are identical.

8. We note that Figure 2A, 3A and 4 in your submission contain copyrighted images. All PLOS content is published under the Creative Commons Attribution License (CC BY 4.0), which means that the manuscript, images, and Supporting Information files will be freely available online, and any third party is permitted to access, download, copy, distribute, and use these materials in any way, even commercially, with proper attribution. For more information, see our copyright guidelines: http://journals.plos.org/plosone/s/licenses-and-copyright.

a. You may seek permission from the original copyright holder of Figure 2A, 3A and 4 to publish the content specifically under the CC BY 4.0 license. 

Reviewers' comments:

Reviewer's Responses to Questions

**Comments to the Author**

1. Is the manuscript technically sound, and do the data support the conclusions?

Reviewer #1: Yes

Reviewer #2: Yes

2. Has the statistical analysis been performed appropriately and rigorously? 

Reviewer #1: Yes

Reviewer #2: Yes

3. Have the authors made all data underlying the findings in their manuscript fully available?

Reviewer #1: Yes

Reviewer #2: Yes

4. Is the manuscript presented in an intelligible fashion and written in standard English?

Reviewer #1: Yes

Reviewer #2: Yes

5. Review Comments to the Author

Reviewer #1: The manuscript entitled “Repressing miR-23a promotes the transdifferentiation of pancreatic α cells to β cells via negatively regulating the expression of SDF-1α” reveals a new mechanism of SDF-1α in pancreatic α-to-β cell transdifferentiation and suggest that miR-23a is a potential target with broad applicability for the treatment of T1D. The manuscript needs minor revision before it can be accepted for publication:

1. The author should improve the english grammer and make the sentences grammatically correct.

2. The authors need to add a few more recent references relevant to the work.

3. The authors need to organize the work well.

Reviewer #2: The articletitled "Repressing miR-23a promotes the pancreatic α-to-β cells transdifferentiation via SDF-1α" authored by Lang et al. underscores the potential significance of miR-23a as a promising therapeutic target in the context of regenerating pancreatic β cells from α cells. This research aims to enhance insulin production, ultimately offering a novel avenue for treating Type 1 Diabetes (T1D). Before considering this article for publication in the journal, certain issues require attention and resolution, as outlined below.

1. The authors mentioned that mice were considered to have Type 1 Diabetes (T1D) when their blood glucose exceeded 200 mg/dl. However, it is essential to clarify whether this measurement is based on fasting or random blood glucose levels.

2. The authors subjected primary islets to very high glucose levels for an extended period. It would be helpful to provide a rationale for this choice of duration and glucose concentration in the study.

3. The TEM images of exosomes isolated from Min6 cells in Fig. 2A appear to be of poor quality. It is advisable to replace these images with higher-quality representations. Also, please include TEM images of the pancreas or the islets in the treated and untreated groups.

4. In the Oral Glucose Tolerance Test (OGTT) experiment, the blood glucose level at 0 minutes (after 14 hours of fasting) was within the normal range, not in the diabetic range. This discrepancy should be explained in the article.

5. Please provide the beta cell mass values for both the untreated and treated groups of mice, as this information is crucial for understanding the impact of the treatment.

6. How would the treatment affect the diabetes progression it was carried out earlier in the mice, for example in the prediabetes stages? Please comment on this or include data if already collected.

7. While insulin levels increased, and the mice showed improved glucose tolerance upon miR23a treatment, it is important to disclose whether there were any changes in body weight within the treated group of mice.

8. Specify the fold change values for dedifferentiation markers such as Aldh1, Neurog3, and any others mentioned in the study.

9. The authors should examine MafA, Pdx1, Neurod1, Urocortin and Ero1-beta in the alpha cells that are transdifferentiated to beta cells. This would add emphasis to the beta cell identity in after α-to-β cell transdifferentiation. On a related note, XBP1, which is linked to Unfolded Protein Response (UPR) events in cells, has been suggested to maintain beta cell identity and function under metabolic stress (in diabetes). The authors should consider analyzing the status/expression of XBP1 in their models in this study.

10. Please include a brief discussion on epigenetic factors that may influence α-to-β cell transdifferentiation.

11. The Immunohistochemistry (IHC) figure for insulin and glucagon are not convincing, as the images for insulin and glucagon appear to be from different pancreatic sections or islets. Please provide accurate and representative immunofluorescence images.

12. In Fig. 2D, where levels of insulin and glucagon are shown using ELISA assays, it would be better to also present this data using Western blot analysis for a more comprehensive assessment. Good

6. PLOS authors have the option to publish the peer review history of their article (what does this mean?). If published, this will include your full peer review and any attached files.

Reviewer #1: **Yes: **Sidra Islam

Reviewer #2: **Yes: **Maroof Alam

---

## [Author Response · Author response to Decision Letter 0]

26 Jan 2024

Reply to Reviewer #1

Reviewer #1: The manuscript entitled “Repressing miR-23a promotes the transdifferentiation of pancreatic α cells to β cells via negatively regulating the expression of SDF-1α” reveals a new mechanism of SDF-1α in pancreatic α-to-β cell transdifferentiation and suggest that miR-23a is a potential target with broad applicability for the treatment of T1D. The manuscript needs minor revision before it can be accepted for publication:

1.The author should improve the english grammer and make the sentences grammatically correct.

Reply: We have thoroughly revised the manuscript to ensure grammatical accuracy and clarity. We have also carefully proofread the entire manuscript to rectify any typographical errors.

2.The authors need to add a few more recent references relevant to the work.

Reply: Thanks for your advice. We apologize for the oversight in not including sufficient recent references. We have conducted a thorough literature review and have identified several relevant recent studies focusing on the regulatory role of SDF-1α and epigenetic foctors in pancreatic cell transdifferentiation. We have incorporated these references (ref. #21 and #29) into the revised version of the manuscript to enhance the discussion and provide a more comprehensive review of the current research landscape.

3. The authors need to organize the work well.

Reply: We acknowledge your comment regarding the organization of the manuscript. We have restructured the content to improve the overall flow and clarity. Specifically, we have rearranged the results and figures (Figure 2 and 3) to each section to enhance readability and facilitate a logical progression of ideas. Our objective was to ensure that the key findings are presented in a clear and coherent manner.

Reply to Reviewer #2

Reviewer #2: The articletitled "Repressing miR-23a promotes the pancreatic α-to-β cells transdifferentiation via SDF-1α" authored by Lang et al. underscores the potential significance of miR-23a as a promising therapeutic target in the context of regenerating pancreatic β cells from α cells. This research aims to enhance insulin production, ultimately offering a novel avenue for treating Type 1 Diabetes (T1D). Before considering this article for publication in the journal, certain issues require attention and resolution, as outlined below.

1.The authors mentioned that mice were considered to have Type 1 Diabetes (T1D) when their blood glucose exceeded 200 mg/dl. However, it is essential to clarify whether this measurement is based on fasting or random blood glucose levels.

Reply: We apologize for the lack of clarity in our manuscript. The blood glucose levels exceeding 200 mg/dl, indicating Type 1 Diabetes (T1D), were based on random blood glucose levels. We have revised the manuscript to clarify this point and ensure accuracy (Line 218).

2.The authors subjected primary islets to very high glucose levels for an extended period. It would be helpful to provide a rationale for this choice of duration and glucose concentration in the study.

Reply: We appreciate your request for a rationale behind the choice of duration and glucose concentration in our study. The primary islets were subjected to high glucose levels for an extended period to simulate the chronic hyperglycemic condition seen in T1D. This prolonged exposure aimed to mimic the impact of sustained hyperglycemia on islet function and viability during the transdifferentiation process. We selected the duration of exposure and glucose concentration based on existing literature and prior experimental evidence, which has shown that chronic exposure to high glucose concentrations can lead to impaired insulin secretion in pancreatic islets [1]. Thus, we chose to expose the islets to high glucose levels for an extended period to closely mimic the chronic hyperglycemia. Nonetheless, we understand the importance of providing a clearer explanation, and we have included additional details and the related reference in the revised manuscript to justify our choice (Line 271-273).

3.The TEM images of exosomes isolated from Min6 cells in Fig. 2A appear to be of poor quality. It is advisable to replace these images with higher-quality representations. Also, please include TEM images of the pancreas or the islets in the treated and untreated groups.

Reply: Thank you for your valuable feedback. We acknowledge your concerns about the quality of TEM images of exosomes presented in Fig. 2A. To address this, we have replaced the original images with higher-quality representations.

Regarding your suggestion for including TEM images of the pancreas or the islets in the treated and untreated groups, we would like to kindly mention that our study focused primarily on the isolation and characterization of exosomes from Min6 cells. Therefore, we did not collect TEM images specifically for the pancreas or the islets in the treated and untreated groups. Our aim was to investigate the characteristics and effects of the exosomes themselves.

Nevertheless, we acknowledge the importance of examining the structural changes within the pancreas or the islets to further support our findings. In future studies, we will consider incorporating such observations to provide a more comprehensive understanding of the effects of the exosomes on the pancreatic tissue.

We appreciate your understanding in this matter and once again want to thank you for your valuable advice. With the improved TEM images of the exosomes, we believe our manuscript will provide valuable insights into the characterization and potential functions of exosomes from Min6 cells.

4.In the Oral Glucose Tolerance Test (OGTT) experiment, the blood glucose level at 0 minutes (after 14 hours of fasting) was within the normal range, not in the diabetic range. This discrepancy should be explained in the article.

Reply: Thanks for your careful advice. We appreciate your observation regarding the blood glucose level at 0 minutes during the Oral Glucose Tolerance Test (OGTT) experiment, where it was noted that the blood glucose level was within the normal range instead of the diabetic range. We understand that this discrepancy should be addressed and clearly explained in the article.

Upon careful consideration, we have thoroughly re-evaluated our experimental procedures and data analysis. We realized that there was an error in the initial data interpretation. We inadvertently misinterpreted the blood glucose levels at 0 minutes, resulting in the incorrect categorization of the subjects as non-diabetic instead of diabetic.(Figure 3C)

In order to rectify this mistake and accurately represent our findings, we have re-analyzed the data and recalculated the blood glucose levels at 0 minutes. We would like to assure you that the revised analysis now correctly reflects the diabetic status of the participants at the start of the OGTT experiment, with blood glucose levels consistent with a state of fasting-induced hyperglycemia.

We deeply apologize for the oversight in our initial interpretation and appreciate your keen observation, which has allowed us to correct this error. The revised manuscript will provide a clear explanation of the revised data analysis and will accurately reflect the diabetic status of the subjects.

5.Please provide the beta cell mass values for both the untreated and treated groups of mice, as this information is crucial for understanding the impact of the treatment.

Reply: Thanks for your professional suggestion. We agree that providing beta cell mass values for both the untreated and treated groups is crucial for understanding the impact of the treatment. In the revised manuscript, we have included the beta cell mass data along with appropriate statistical analyses to offer a comprehensive assessment (Figure 3B, Line 301).

6.How would the treatment affect the diabetes progression it was carried out earlier in the mice, for example in the prediabetes stages? Please comment on this or include data if already collected.

Reply: We appreciate your suggestion to investigate the effects of treatment in prediabetic stages. It has been observed that miR-23 is significantly increased in individuals with Type 1 Diabetes (T1D), particularly in those with a C-peptide level below 200 pmol/L, indicating its potential as a biomarker for severe T1D. Our results have demonstrated that suppression of miR-23a promotes α-to-β like cell transition both in vitro and in vivo. However, we did not identify the exact role of miR-23a in the progression of diabetes, and further investigation is warranted to elucidate this matter. To address this, relevant commentary has been included in the DISCUSSION section, specifically in lines 369-371.

7.While insulin levels increased, and the mice showed improved glucose tolerance upon miR23a treatment, it is important to disclose whether there were any changes in body weight within the treated group of mice.

Reply: We apologize for not specifically mentioning changes in body weight within the treated group of mice. Upon further analysis, we found no significant changes in body weight following miR-23a treatment. We have now included this information in the Supplementary Figure 1 and the revised manuscript (Line 302-303).

8.Specify the fold change values for dedifferentiation markers such as Aldh1, Neurog3, and any others mentioned in the study.

Reply: Thanks for your professional advice. We have specified the fold change values for dedifferentiation markers such as Aldh1, Neurog3 in the study (Figure 3I). These additional data enhance the quantitative assessment of the dedifferentiation process and strengthen our findings.

9. The authors should examine MafA, Pdx1, Neurod1, Urocortin and Ero1-beta in the alpha cells that are transdifferentiated to beta cells. This would add emphasis to the beta cell identity in after α-to-β cell transdifferentiation. On a related note, XBP1, which is linked to Unfolded Protein Response (UPR) events in cells, has been suggested to maintain beta cell identity and function under metabolic stress (in diabetes). The authors should consider analyzing the status/expression of XBP1 in their models in this study.

Reply: Thanks for your constructive advice. We acknowledge the importance of examining additional markers that emphasize beta cell identity after α-to-β cell transdifferentiation. In response to your suggestion, we have included the analysis of MafA, Pdx1, NeuroD1, Urocortin, Ero1-beta, and XBP1 in the alpha cells that undergo transdifferentiation to beta cells (Figure 2F and Figure 3H). These analyses provide valuable insights into the maintenance and functionality of the transdifferentiated beta cells in our study.

10. Please include a brief discussion on epigenetic factors that may influence α-to-β cell transdifferentiation.

Reply: We appreciate your suggestion to include a brief discussion on epigenetic factors that may influence α-to-β cell transdifferentiation. In the revised manuscript, we have included a discussing the potential influence of epigenetic factors on this process (Line 372-377). This addition contributes to a more thorough understanding of the regulatory mechanisms involved.

11.The Immunohistochemistry (IHC) figure for insulin and glucagon are not convincing, as the images for insulin and glucagon appear to be from different pancreatic sections or islets. Please provide accurate and representative immunofluorescence images.

Reply: We apologize for any confusion caused by the previous IHC figure for insulin and glucagon, where the images appeared to be from different pancreatic sections or islets. We have now replaced these images with accurate and representative immunofluorescence images which clearly demonstrate the localization of insulin and glucagon within the same pancreatic sections or islets.

12. In Fig. 2D, where levels of insulin and glucagon are shown using ELISA assays, it would be better to also present this data using Western blot analysis for a more comprehensive assessment. Good

Reply: We appreciate your suggestion to complement the ELISA assay data with Western blot analysis for a more comprehensive assessment of insulin and glucagon levels in previous version of Fig. 2D. In response, we have performed Western blot analysis in addition to the ELISA assay, and the data have been presented in the revised manuscript (Figure 2F).

References

1. Castex F, Leroy J, Broca C, Mezghenna K, Duranton F, Lavallard V, et al. Differential sensitivity of human islets from obese versus lean donors to chronic high glucose or palmitate. Journal of diabetes. 2020;12(7):532-41.

---

## [Decision Letter · Decision Letter 1]

16 Feb 2024

Repressing miR-23a promotes the transdifferentiation of pancreatic α cells to β cells via negatively regulating the expression of SDF-1α.

PONE-D-23-30492R1

Dear Dr. kang,

We’re pleased to inform you that your manuscript has been judged scientifically suitable for publication and will be formally accepted for publication once it meets all outstanding technical requirements.

Kind regards,

Syed M. Faisal, Ph.D.

Academic Editor

PLOS ONE

Additional Editor Comments (optional):

Reviewers' comments:

Reviewer's Responses to Questions

**Comments to the Author**

1. If the authors have adequately addressed your comments raised in a previous round of review and you feel that this manuscript is now acceptable for publication, you may indicate that here to bypass the “Comments to the Author” section, enter your conflict of interest statement in the “Confidential to Editor” section, and submit your "Accept" recommendation.

Reviewer #2: All comments have been addressed

2. Is the manuscript technically sound, and do the data support the conclusions?

Reviewer #2: Yes

3. Has the statistical analysis been performed appropriately and rigorously? 

Reviewer #2: Yes

4. Have the authors made all data underlying the findings in their manuscript fully available?

Reviewer #2: (No Response)

5. Is the manuscript presented in an intelligible fashion and written in standard English?

Reviewer #2: Yes

6. Review Comments to the Author

Reviewer #2: (No Response)

7. PLOS authors have the option to publish the peer review history of their article (what does this mean?). If published, this will include your full peer review and any attached files.

Reviewer #2: **Yes: **Maroof Alam

---

## [Editor Report · Acceptance letter]

11 Mar 2024

PONE-D-23-30492R1 

PLOS ONE

Dear Dr. kang, 

I'm pleased to inform you that your manuscript has been deemed suitable for publication in PLOS ONE. Congratulations! Your manuscript is now being handed over to our production team.

Kind regards, 

on behalf of

Dr. Syed M. Faisal 

Academic Editor

PLOS ONE